# Metal-organic framework patterns and membranes with heterogeneous pores for flow-assisted switchable separations

Guan-Young Jeong[1], Ajay K. Singh[1], Min-Gyu Kim[2], Ki-Won Gyak [1], UnJin Ryu[3], Kyung Min Choi[3] & Dong-Pyo Kim[1]

Porous metal-organic-frameworks (MOFs) are attractive materials for gas storage, separations, and catalytic reactions. A challenge exists, however, on how to introduce larger pores juxtaposed with the inherent micropores in different forms of MOFs, which would enable new functions and applications. Here we report the formation of heterogeneous pores within MOF particles, patterns, and membranes, using a discriminate etching chemistry, called silver-catalyzed decarboxylation. The heterogeneous pores are formed, even in highly stable MOFs, without altering the original structure. A decarboxylated MOF membrane is shown to have pH-responsive switchable selectivity for the flow-assisted separation of similarly sized proteins. We envision that our method will allow the use of heterogeneous pores for massive transfer and separation of complex and large molecules, and that the capability for patterning and positioning heterogeneous MOF films on diverse substrates bodes well for various energy and electronic device applications.

[1] Center of Intelligent Microprocess for Pharmaceutical Synthesis, Department of Chemical Engineering, POSTECH (Pohang University of Science and Technology), Pohang 37673, Korea. [2] Beamline Research Division, Pohang Accelerator Laboratory (PAL), POSTECH (Pohang University of Science and Technology), Pohang 37673, Korea. [3] Department of Chemical and Biological Engineering and Institute of Advanced Materials & Systems, Sookmyung Women's University, 100 Cheongpa-ro 47 gil, Yongsan-gu, Seoul 04310, Korea. These author contribute equally: Guan-Young Jeong, Ajay K. Singh Correspondence and requests for materials should be addressed to K.M.C. (email: kmchoi@sm.ac.kr) or to D.-P.K. (email: dpkim@postech.ac.kr)

Homogeneous and microporous spaces periodically arranged in metal-organic frameworks (MOFs) have allowed access of guest molecules smaller than the pore size, which has been utilized for storage, separation, or conversion applications[1–4]. However, a challenge still exists on how to use microporous MOFs for complex and large organic, inorganic, and biological molecules[5]. We believe that introduction of larger pores juxtaposed with the micropores of MOF in various forms of MOF particles, patterns, and membranes is needed for high level of functions and applications[6]. For instance, meso- or macro-pores within MOFs provide fast pathways to enhance mass transfer or the space for storing and separating large and complex molecules, while the intrinsic micropores contribute to the high surface area and host–guest interactions[2,6,7].

Various approaches have been taken to create relatively large pores in MOFs, including longer building units, modulators, imperfect crystallization, and templates[8–12]. In particular, the post-treatment methods of sacrificing the micropores within MOFs to yield larger pores usually required a harsh and tedious process[13]. The severe weight loss incurred during the etching process led to serious problems of maintaining the original morphology in powders, patterns and membranes[14]. These problems can cause indiscriminate damage to their supporting substrates as well as MOFs themselves[15]. Although a study producing a mesoporous MOFs film using electrochemical procedure was reported[16], a practical and versatile method is still needed that is applicable to any MOFs regardless of their form.

The approach we took for producing heterogeneous pore structures in MOFs is outlined in Fig. 1. It is based on a discriminate etching chemistry, named silver-catalyzed decarboxylation, under mild conditions. The decarboxylation involves a chemical reaction that removes a carboxyl group of organic linkers in MOFs, and then makes meso- and macro-pores while preserving the overall MOF structure. The sulfate ion generated by the silver catalyst in the etching solution is responsible for the decarboxylation and thus the production of the mesopores. This decarboxylated MOF is hereafter denoted as MOF-d. The effective formation of heterogeneous pores is demonstrated for a representative Cu-based MOF (HKUST-1) and then for Zr- (UiO-66), Eu- (Eu-MOF), Al-based MOFs [MIL-100(Al) and MIL-53(Al)], Cr- [MIL-101(Cr)] and Fe- [MIL-100(Fe)]. These MOFs are known to have exceptionally high structural stability and therefore difficult to make larger pores without altering the original appearance. This decarboxylation approach is equally effective not only for MOF particles but also for MOF patterns and MOF membranes while retaining the integrity, as illustrated in Fig. 1.

## Results

**Decarboxylation of MOF particles**. To check the validity of the decarboxylation strategy, pristine HKUST-1 was first synthesized by solvothermal reaction of $Cu(NO_3)_2$ with $H_3BTC$ (1,3,5-benzenetricarboxylic acid) in an ethanol–water mixture[17]. The resulting HKUST-1 particles were then mixed with $AgNO_3$ and $K_2S_2O_8$ of equal mass ratio in acetonitrile (ACN). The mixture was transferred into an autoclave reactor at 120 °C for the decarboxylation reaction.

Fig. 2 shows the results obtained by scanning electron microscopy (SEM), powder X-ray diffraction (PXRD), and $N_2$ sorption measurement on a series of HKUST-1-dX samples where X is reaction time in minutes. Pristine HKUST-1 sample shows a very smooth surface with no observable pores (Supplementary Figure 1). The mesopores were created after 20 min of decarboxylation (Fig. 2a) and started forming larger pores with sponge-like morphology upon extending the reaction time to 60 min (Fig. 2b, c). The fact that the diffraction peaks of HKUST-1-d20, -d40, and -d60 all coincide (Fig. 2d) is a strong indication that the chemical structure and crystallinity remains unchanged except slight peak broadening in -d60 sample even after the formation of larger pores[13,14]. Note that there were no residues of Ag and K ion after the post-washing process, as seen by energy-dispersive X-ray spectroscopy in SEM (Supplementary Figure 2) and ICP analysis. When the pore formation process was extended up to 24 h for 170 °C, the samples lost their crystallinity and crystal shape as observed by PXRD and SEM (Supplementary Figure 3).

The presence of meso- and macro-pores in bulk HKUST-1-d20, -d40, and -d60 samples was confirmed by measurement of $N_2$ sorption isotherm (Fig. 2e). The pristine HKUST-1 sample showed typical type I isotherm with no hysteresis during desorption. However, the HKUST-1-d20, -d40, and -d60 samples exhibited a typical mesoporous hysteresis loop over the relative pressure range of $0.5 < P/P_0 < 0.95$ with type IV isotherm behavior. The pore-size distribution profile, calculated by the Barrett–Joyner–Halenda (BJH) method, showed that the

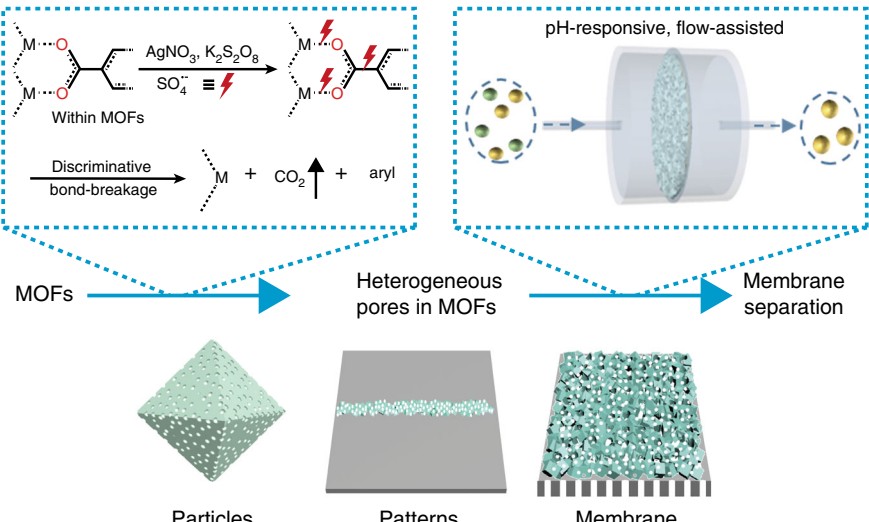

**Fig. 1** Heterogeneous MOFs via decarboxylation process. The process can be applied to obtain heterogeneous pores in different forms of MOFs such as particles, patterns, and membranes

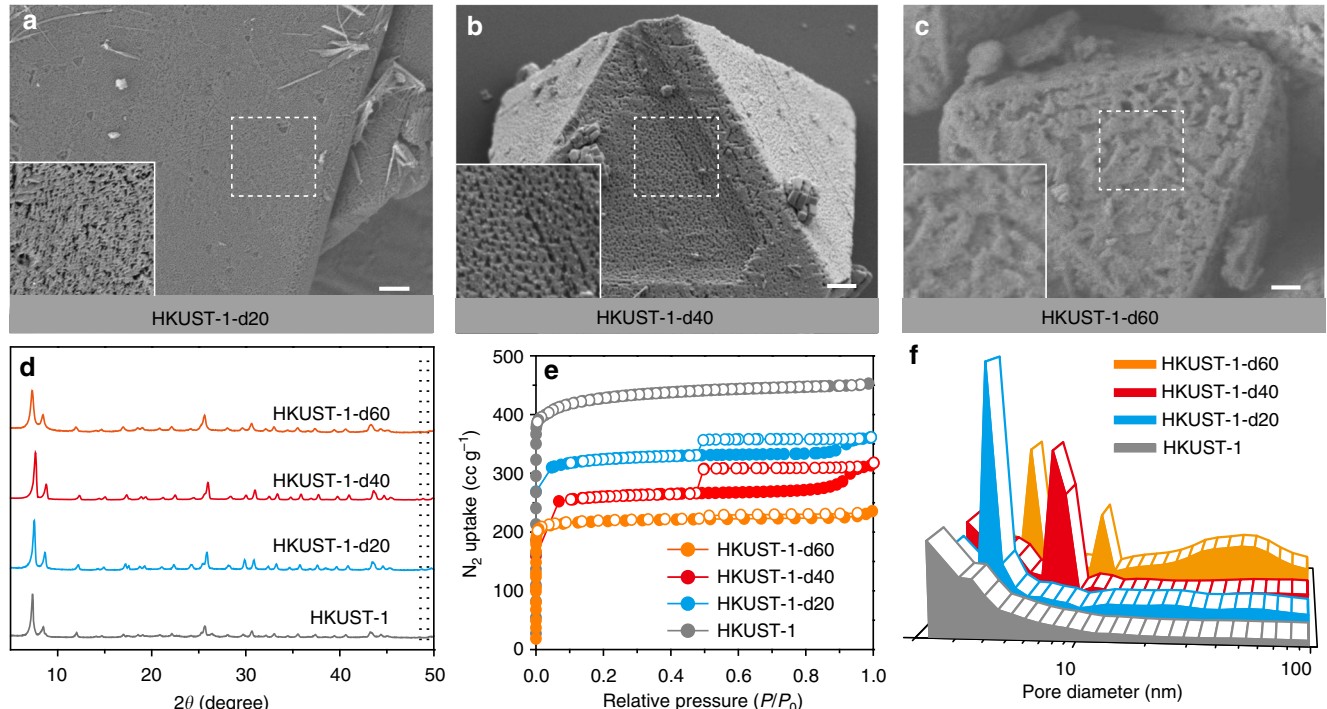

**Fig. 2** Development of the meso- and macro-pores within HKUST-1 via decarboxylation at 120 °C. The product of the decarboxylation process was examined every 20 min up to 60 min of residence time. SEM images of **a** HKUST-1-d20, **b** HKUST-1-d40, and **c** HKUST-1-d60 (scale bar is 1 μm), **d** X-ray diffraction patterns, **e** N$_2$ sorption isotherms, and **f** BJH pore-size distribution of HKUST-1, HKUST-1-d20, -d40, and -d60

pore size was progressively getting larger as the decarboxylation time increases (Fig. 2f). After 60 min of reaction, macro-pores started forming in the size range between 50 and 100 nm. The surface area ($S_{BET}$) gradually decreased from initial value of 1690 to 1191 m$^2$ g$^{-1}$ as the decarboxylation time increased to 60 min, while the total specific pore volume ($V_t$) increased from the initial value of 0.36 to 0.73 cm$^3$ g$^{-1}$ (Supplementary Table 1).

The micropore volume of HKUST-1-d40 remained at ~70% of the pristine HKUST-1. Considering that other treatments based on water and plasma led to ~30% and ~50% of the original micropore volume remaining after the treatment, micropores are most efficiently preserved by the decarboxylation treatment in the process of creating meso- and macro-pores. Moreover, the micropore volume of 0.11 cm$^3$ g$^{-1}$ was transformed into a mesoporous volume of 0.43 cm$^3$ g$^{-1}$ such that the ratio of mesopore gain to micropore loss is 3.88. This value is significantly higher than those reported for other treatment methods (1.80 in the case of water treatment, 1.11 of H$_3$PO$_4$, and 1.18 of HCl; Supplementary Table 2), revealing the high efficiency of generating most heterogenous pores with least consumption of micropores. When the etching time was extended to 90 min (HKUST-1-d90), macro-pores in excess of 50 nm formed, as determined by SEM image, PXRD and BJH pore-size distribution (Supplementary Figure 4a–c and Supplementary Table 1). It is worth nothing that the HKUST-1-d60 particles retained the initial morphology, despite the generation of meso- and macro-pores in the framework. These results successfully demonstrated the validity of the decarboxylation strategy as an efficient etching chemistry of MOFs to generate meso- and macroporous HKUST-1 within 1 h of reaction time.

Creating meso- and macro-pores in MOFs with strong chemical bonding is a difficult task because it is not easy to break the bonding. To test the robustness of the decarboxylation approach, we applied it to chemically robust MOFs of UiO-66, Eu-MOF, MIL-100(Al), and MIL-53(Al). All MOFs tested were successfully transformed to meso- and macroporous structure to give UiO-66-d40, Eu-MOF-d40, MIL-100(Al)-d40, and MIL-53 (Al)-d40 when a slightly stronger decarboxylation condition of 150 °C was applied. For instance, decarboxylation of UiO-66 at 120 °C led to a low etching efficiency (Supplementary Figure 5) whereas an effective etching activity was observed at 150 °C (Supplementary Figure 6). The surface area of the decarboxylated UiO-66 at 150 °C for 40 min decreased from 1720 to 1226 m$^2$ g$^{-1}$ while the total pore volume increased from 0.77 to 1.40 cm$^3$ g$^{-1}$. The micropore volume of UiO-66-d40 was preserved at 74% of the pristine UiO-66 and the ratio of mesopore gain to micropore loss was 4.70.

The presence of mesopores is clearly seen in the TEM images of Eu-MOF-d40 (Supplementary Figure 7a, b), and the SEM images of MIL-100(Al)-d40 (Supplementary Figure 8a, b) and MIL-53-d40 (Supplementary Figure 9a, b) that were obtained by decarboxylation at 150 °C for 40 min. Quantitative analysis results based on nitrogen sorption are summarized in Supplementary Table 3. The changes in X-ray diffraction, surface area, total pore volume, micropore volume, and the ratio of mesopore gain to micropore loss are given in Supplementary Figure 7c, d for Eu-MOF-d40, in Supplementary Figure 8c, d for MIL-100(Al)-d40, and in Supplementary Figure 9c, d for MIL-53(Al)-d40, summarizing in Supplementary Table 3. The formation of mesopores into extremely stable MOFs, such as MIL-101(Cr) and MIL-100(Fe), was also confirmed by TEM and PXRD for MIL-101(Cr)-d40 (Supplementary Figure 10a–c) and SEM and PXRD for MIL-100(Fe)-d40 (Supplementary Figure 11a–c). Not surprisingly, zeolitic imidazole frameworks that do not have the carboxylic ligand showed no change in TEM image as well as PXRD and N$_2$ sorption isotherm even after decarboxylation at 150 °C for 40 min (Supplementary Figure 12).

To gain a mechanistic understanding of the decarboxylation reaction in HKUST-1, analyses by ex-situ XANES and EXAFS were conducted. The decarboxylation is known to break C–C

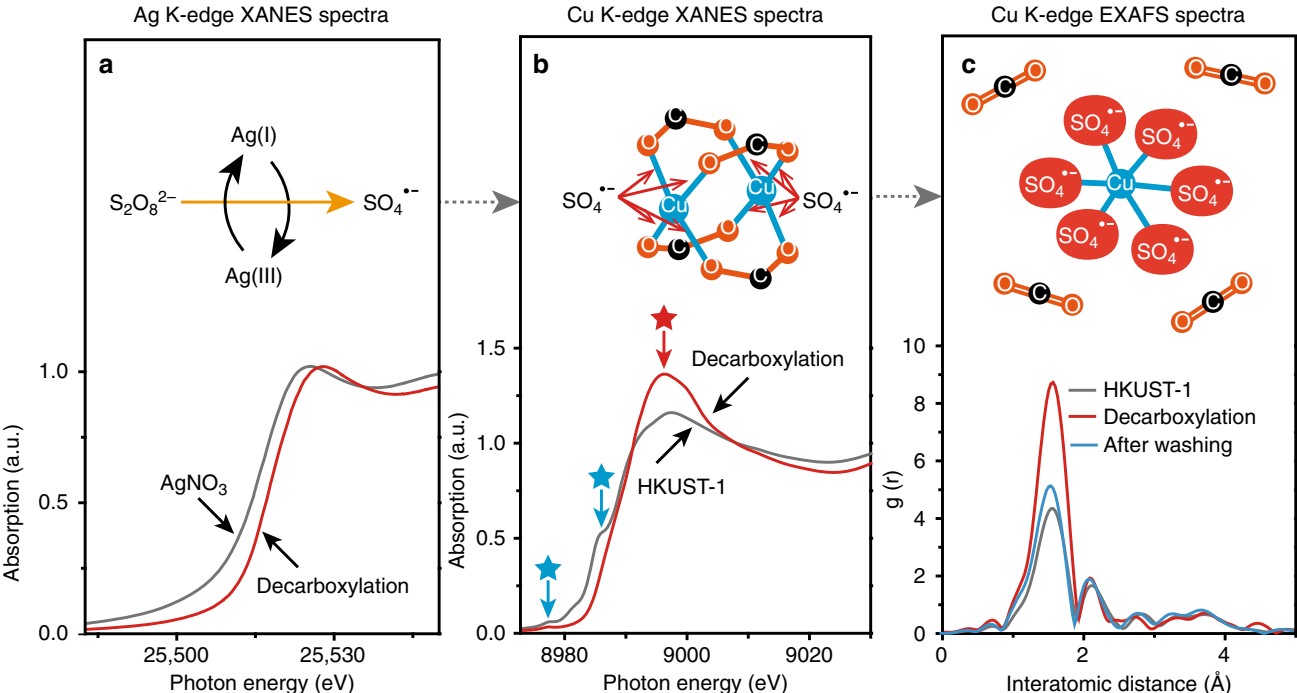

**Fig. 3** XANES and EXAFS spectra measured during decarboxylation at 120 °C. **a** Change in Ag K-edge XANES spectra for AgNO₃ and **b** change in Cu K-edge XANES spectra, **c** change in Cu K-edge EXAFS spectra before and after the decarboxylation and after washing process

bond to free $CO_2$ from the carboxylic group[18], but it is not known whether it attacks M–O (M represents metal) bond as well that exists in MOFs in the process of yielding meso- and macro-pores. To acquire the bond information, the decarboxylation process and metal coordination environment inside MOFs was monitored by XANES and EXAFS. The role of the Ag catalyst (AgNO₃) during decarboxylation was observed by analyzing Ag K-edge XANES spectra (Fig. 3a). The chemical state of AgNO₃ was initially Ag(I) as confirmed by the peak inflection energy (25515.5 eV) in XANES spectra[19]. During decarboxylation in the presence of HKUST-1, its chemical state was changed to Ag (III) as evidenced by the peak inflection energy shift to 25518.7 eV (Fig. 3a)[20–22], which indicates that AgNO₃ oxidatively transfers two electrons for the generation of $SO_4^{\bullet-}$ (ref. [23]). The existence of Ag(III) catalyst intermediately suggests fast decarboxylation reaction with the help of $SO_4^{\bullet-}$.

The change in the chemical state of Cu(II) existing in metal oxide parts of HKUST-1 was observed by Cu K-edge XANES spectra (Fig. 3b). The Cu(II) showed a shoulder edge at 8990 eV (red star in Fig. 3b) and two characteristic pre-edge peaks at ca. 8976 and ca. 8986 eV (blue stars in Fig. 3b), suggesting a square planar geometry of Cu center (shown in the upper part of Fig. 3b)[24]. In decarboxylation reaction, these distinguished shoulder peaks completely disappear, which represents that Cu(II) in HKUST-1 is partially oxidized by $SO_4^{\bullet-}$ and formed the octahedral geometry of Cu center. For the Cu oxide parts in HKUST-1, Cu K-edge EXAFS spectra were used to investigate the coordination change of Cu in the process of decarboxylation. The coordination number of Cu was measured before and after the decarboxylation reaction, and after final washing process. The central Cu has first shell signal (at ca. 1.5 Å, not corrected in phase) corresponding to Cu–O bonds with coordination number of 4 in square planar geometry (Fig. 3c)[24]. After decarboxylation, Cu K-shell signal intensity increased approximately 1.5 times, and the coordination number around central Cu changed to 6 in octahedral geometry. After washing with fresh water, the coordination number decreased to 4, reverting to the original square planar geometry (Fig. 3c). These results indicate

that the $SO_4^{\bullet-}$ was adjacent to the microporous structure and cleaved the carboxylate group by decarboxylation chemistry to form highly oxidized Cu-sulfate complexes. This process leads to the partial removal of linkers by breaking not only C–C bonds but also M–O bonds in the HKUST-1. The washing removes the Cu-sulfate complexes and the remaining catalysts from the framework, which results in the formation of larger pores.

The remained carboxyl ligand of HKUST-1-d20, 40, 60, and 90 samples was quantitatively monitored by measuring $CO_2^-$ stretching peak of Infrared (IR) spectrum (Supplementary Figure 13a). The integrated intensities of a weak asymmetrical stretching, $v_{as}(CO_2^-)$, at 1400−1500 cm⁻¹ and a relatively strong symmetrical stretching, $v_s(CO_2^-)$, at 1300−1400 cm⁻¹ region, were progressively decreased with 7% and 11% for HKUST-1-d20 and -d40, then severely reduced with 31% and 43% upon extended decarboxylation reaction to 60 and 90 min, respectively (Supplementary Figure 13). Moreover, the supernatant solutions obtained from HKUST-1-d20, 40, 60 samples were analyzed by gas chromatography mass spectroscopy (Supplementary Figure 14). The volatile benzene peak was commonly found with ACN and dichloromethane (DCM, as activating agent). In particular, the HKUST-1-d40 and -d60 samples showed obvious presence of benzene in the solutions, which is clearly evident for decarboxylation reaction occurred in H₃BTC (1,3,5-benzenetricarboxylic acid) ligand as dominant chemistry of decarboxylation.

**Decarboxylation of MOF patterns.** With the decarboxylation approach verified, we looked into the possibility of tailoring pore structures at desired locations in any form or pattern desired. This capability is highly desirable for device fabrication[22,23], particularly when the tailoring can be accomplished on any substrate while retaining the integrity of the entire MOF patterns. A durable MOFs of UiO-66 was chosen for the demonstration. To enable the MOF growth[24] on desired locations, glass and Si substrate surfaces were modified with thioglycolic acid and polymer substrate surfaces with polydopamine (Supplementary

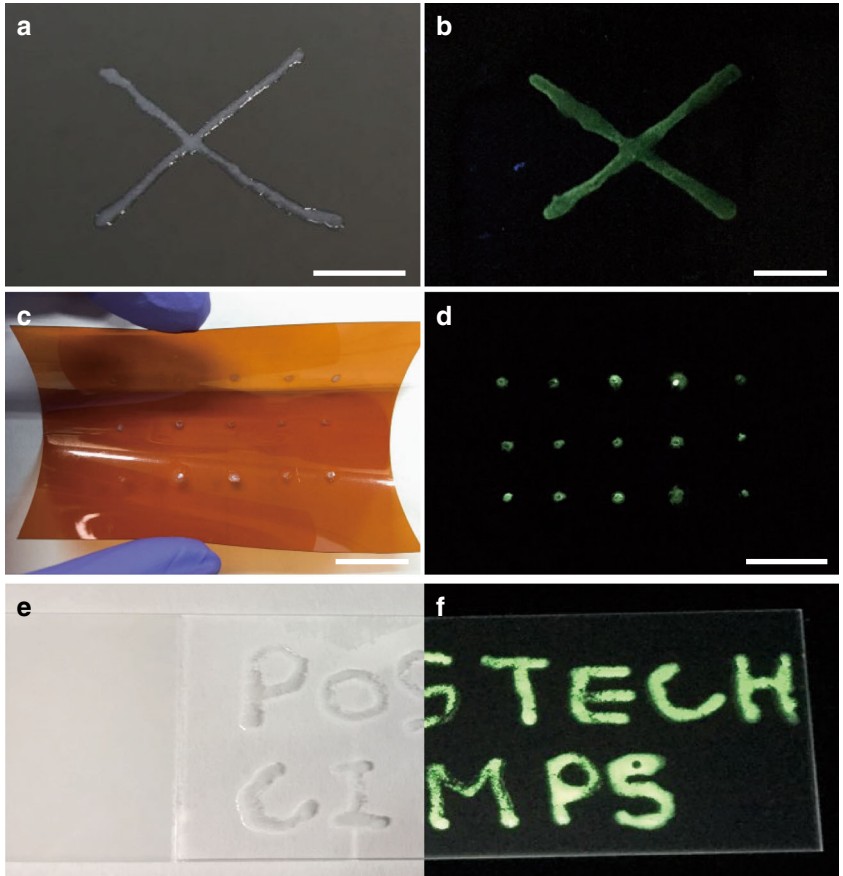

**Fig. 4** Mesoporous MOF patterns. **a** Photo and **b** luminescence images of "X"-shaped UiO-66-d40 pattern on an Si substrate (scale bar is 0.5 cm). **c** Photo and **d** luminescence images of dot-shaped UiO-66-d40 pattern on flexible polyimide film (scale bar is 2 cm). **e** Photo and **f** luminescence images of "POSTECH" patterned UiO-66-d40 on a glass substrate

Figure 15a). This surface treatment enhances the interfacial adherence between the UiO-66 patterns and the substrates by initiating the growth of UiO-66 patterns at the desired location (Supplementary Figure 15b). The UiO-66 patterns underwent the decarboxylation process at 150 °C for 40 min to give UiO-66-d40 patterns to produce meso- and macro-pores within them (Supplementary Figure 15c). The UiO-66 pattern thus grown at the modified surface in the shape of "X" on Si substrate and then decarboxylated into UiO-66-d40 pattern is shown in Fig. 4a. The developed meso- and macro-porosity of the UiO-66-d pattern was visualized by immobilizing a fluorescence dye (20,000 MW of FITC- dextran, 3 nm × 2 nm × 3 nm in size) inside the meso- and macro-pores (Fig. 4b). As the size of the dye is bigger than the micropore, the fluorescence was only found in UiO-66-d40 pattern, and not in the pristine UiO-66 pattern. Dot-patterned UiO-66-d40 was also successfully formed on the polyimide flexible film (Fig. 4c). Initially, UiO-66 was grown on the polydopamine deposited dot-pattern on the polyimide film and then decarboxylated. The meso- and macro-porosity within the dot-patterned UiO-66-d40 was well revealed by the FITC-dextran fluorescence treatment (Fig. 4d). Furthermore, a glass substrate was modified by thioglycolic acid to produce a finger sketched "POSTECH" characters. The image of the synthesized and then decarboxylated UiO-66-d40 pattern is shown in Fig. 4e and its dyed version in Fig. 4f.

The surface of Si, polyimide and glass substrates, and the MOF pattern are well preserved with robust adhesion even after the decarboxylation process. The meso- and macro-pores existing within the UiO-66-d40 pattern were examined by

high-angle annular dark-field scanning transmission electron microscopy (HAADF-STEM) after removing the MOF particles from the substrates (Supplementary Figure 15d, e). The HAADF-STEM image shows a nanocrystalline UiO-66-d40 particle (ca. 450 nm in diameter) containing the enlarged pores that are homogeneously distributed throughout the whole particle (Supplementary Figure 15e). The SEM image indicated that the pores are also developed on the exterior surface of the crystal as well (Supplementary Figure 15d). N$_2$ sorption experiments for UiO-66-d40 samples indicated that the meso- and macro-pores in UiO-66-d40 pattern successfully formed on the desired substrate (Supplementary Figure 16). The X-ray diffraction lines of UiO-66 pattern and UiO-66-d40 pattern are well-corresponding with simulated UiO-66 diffraction. It is believed that the decarboxylation approach makes larger pores with same chemical structure and crystallinity, even in the presence on any substrate.

These results revealed that the direct formation of meso- and macroporous UiO-66 patterns could easily be accomplished with the decarboxylation approach without affecting the integrity of both MOF patterns and substrates, which could not be achieved by previous methods[16]. Therefore, this patterning technology of heterogeneous MOF pores on Si-wafer and glass slide as substrates would be compatible with the conventional lithographic process for fabricating functional chip devices[25]. In particular, the use of polyimide film bodes well for fabricating flexible, wearable electronic devices. It is expected that this discriminate etching approach would enable a broader selection of substrates for various applications[26].

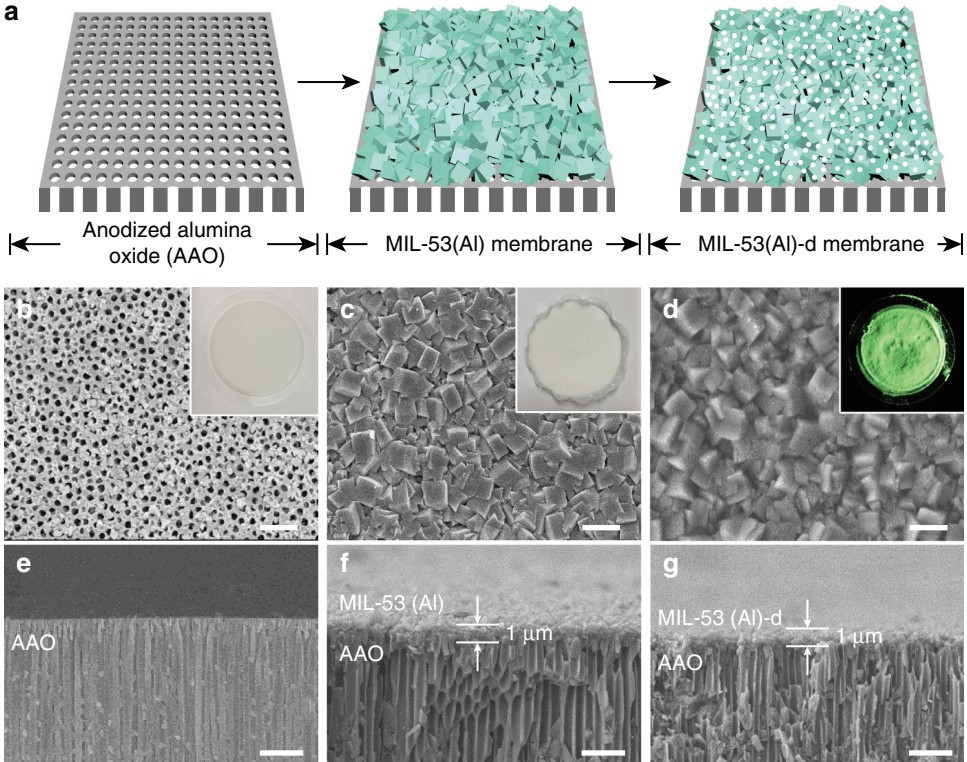

**Fig. 5** Mesoporous MOF membranes. **a** Conceptual illustration of sequential fabrication of MIL-53(Al)-d40 membrane. Top view: SEM micrographs of **b** AAO membrane, **c** MIL-53(Al) membrane, and **d** MIL-53(Al)-d40 membrane. Cross view SEM micrographs of **e** AAO membrane, **f** MIL-53(Al) membrane, and **g** MIL-53(Al)-d40 membrane. **b**, **e** AAO membrane as received, **c**, **f** growth of MIL-53(Al) membrane for 4 h, and **d**, **g** decarboxylated MIL-53(Al)-d40 membrane. The insets: **b** digital photo, **c**, **d** luminescence images with FITC-dextran. **b**–**d** Scale bar is 1 μm; **e**–**g** scale bar is 2 μm

**Decarboxylation of MOF membrane**. In addition to the device applications, the meso- and macroporous MOF membrane with mechano-chemical stability has high potential for massive transfer and separation of complex and large molecules in aqueous mixtures, which has not been possible with vulnerable MOF membranes with hydrolytic instability[27]. The successful fabrication of various MOF-d patterns on diverse substrates that has been demonstrated encouraged us to prepare an MOF-d membrane on a nanoporous AAO (anodic aluminum oxide) substrate, potentially for utilization as a protein separation membrane. Durable MIL-53(Al) was chosen for the purpose with an AAO support (0.02 μm pore, 25 mm diameter). In this MOF growth, AAO plays an important role of providing Al ions for the formation of MIL-53(Al), which makes strong adhesion between them[18]. Then, the MIL-53(Al) film on AAO underwent the decarboxylation process at 150 °C for 40 min to give MIL-53(Al)-d40 membranes, as illustrated in Fig. 5a. Macroporous AAO supports (Fig. 5b, e) was fully covered with MIL-53(Al) membrane with a thickness of 1 μm after 4 h of MIL-53(Al) synthesis (Fig. 5c, f). After decarboxylation, the MIL-53(Al)-d40 membranes maintain its original film morphology and crystal shape (Fig. 5d, g) without any damage to the AAO substrate (Fig. 5e–g).

The existence of mesoporosity in the MIL-53(Al)-d40 membrane was clearly visualized by selectively immobilizing the FITC-dextran (20,000 MW) within meso- and macro-pores as shown in the inset of Fig. 5d. Note that the pristine MIL-53(Al) membrane did not show observable fluorescence due to the absence of the meso-scaled dye that could not penetrate inside micropores. TEM image of the detached MIL-53(Al)-d40 sample from the membrane sample also confirmed the existence of meso- and macro-pores, consistent with the characteristics of the MIL-53(Al)-d40 powder (Supplementary Figure 17a). The retained

crystallinity of MIL-53(Al)-d40 membranes was verified by PXRD measurement (Supplementary Figure 17b). The sharp diffraction peaks at identical position indicate that MIL-53(Al)-d40 membranes have the same structure as MIL-53(Al), while a slightly broad background as observed is attributed to amorphous AAO substrate.

The MIL-53(Al)-d40 on AAO is quite suitable for separating biomolecules as a nanofilter[28]. Although many techniques are available for separation of proteins of different size, separating proteins of similar size is quite another matter[28–30]. Our strategy is to use the meso- and macro-pores as massive pathways for continuous flow-assisted protein movement, while allowing the surrounding micropores absorb charged molecules to give electrostatic attraction/repulsion to the proteins (Fig. 6a). For this purpose, the MIL-53(Al) membrane was synthesized using 2,2′-bipyridine-5,5′-dicarboxylate(bpy) and followed by the decarboxylation reaction. The resulting MIL-53(Al)bpy-d40 membrane was submerged in the iodomethane solution, so that the iodomethane was supposed to attach to the pyridine site and make the micropores positively charged to give MIL-53(Al)bpy$^+$-d40 membrane. This process is known as N-quaternization[31]. The zeta potential of the MIL-53(Al)bpy$^+$-d40 membrane was shifted positively after the N-quaternization (Supplementary Figure 18), which clearly supports the fact that N-quaternization makes the micropores of MIL-53(Al)bpy-d membrane positively charged. The nanofilter apparatus for conducting a continuous flow-assisted protein separation was assembled by mounting the MIL-53(Al)bpy$^+$-d40 membrane into a home-made assembly holder (left part of Fig. 6b). The water flux was measured around 1955 L m$^{-2}$ h$^{-1}$ bar$^{-1}$, with assumption of pressure of 1 bar. A leakage test was performed by pouring a fluorescence dye (100,000 MW, FITC-dextran) solution into the

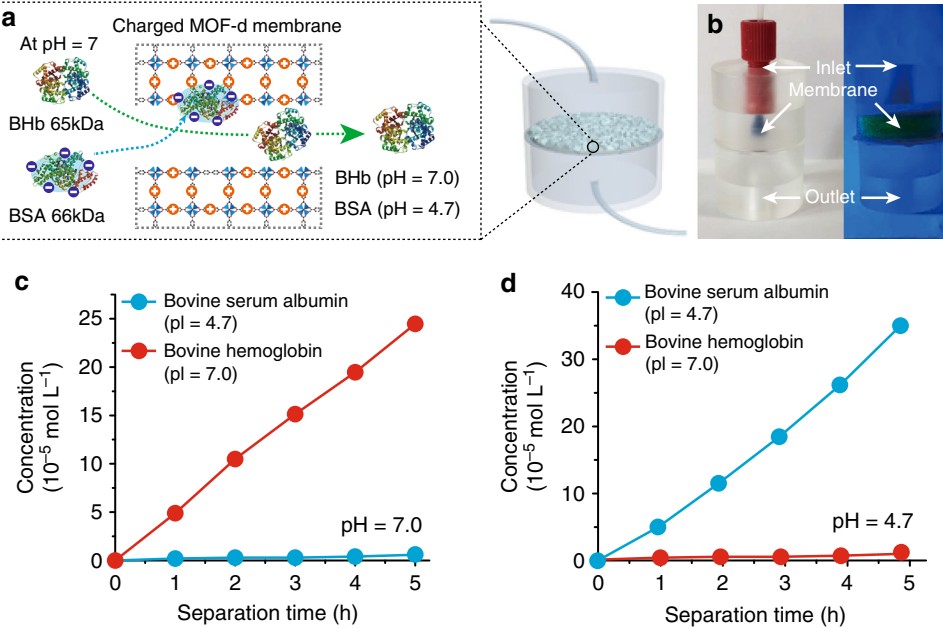

**Fig. 6** Protein separation by mesoporous MOF membranes. **a** Conceptual illustration of continuous flow-assisted protein separation by positively charged MIL-53(Al)bpy-d40 membrane embedded nanofilter device with a pH-responsive switchable selectvity. **b** Separation device composed of inlet, MIL-53(Al)bpy$^+$-d40 membrane and outlet, right side image shows the leakage test results by FITC-dextran (MW: 100,000). **c** Concentration profiles of the filtered proteins as a function of separation time at pH 7.0. **d** Concentration profiles of the filtered proteins as a function of separation time at pH 4.7, analyzed by UV–vis spectra

top reservoir. Even after several hours, no solution leaked into the bottom reservoir as indicated in the right part of Fig. 6b.

Two types of similar-sized proteins, bovine serum albumin (BSA, 66 kDa) and bovine hemoglobin (BHb, 65 kDa), were prepared to evaluate the separation efficiency of MIL-53(Al)-bpy$^+$-d membrane. As the hydrodynamic sizes of BSA and BHb are $14 \times 3.8 \times 3.8$ nm (ellipsoid) and $6.4 \times 5.5 \times 5$ nm (spherical)[30], respectively, the mesopores larger than 5 nm in the nanofilter provide the pathway for the proteins passing through. The isoelectric points (pI) of BSA and BHb are 4.7 and 7.0, respectively, which means that BSA is negatively charged while BHb is neutral at pH 7.0 (Fig. 6a). When pH is changed to 4.7, BSA is neutral and BHb gets positively charged. The protein mixture (concentration at 0.066 mg ml$^{-1}$) adjusted to the desired pH (7.0 and 4.7) by buffer solution was injected by the syringe pump onto the top of the nanofilter at a flow rate of 0.16 ml min$^{-1}$. The UV–vis spectra of the collected sample from the bottom outlet were used for the separation capability (Supplementary Figure 19). When the pH was 7.0, the concentration of BHb gradually increased from $4.9 \times 10^{-5}$ mol l$^{-1}$ after 1 h to $24.47 \times 10^{-5}$ mol l$^{-1}$ after 5 h but negligible BSA was detected even after 5 h (Fig. 6c). The neutral BHb should freely pass through the meso- and macro-pores in MIL-53(Al)bpy$^+$-d membrane while the negatively charged BSA would be trapped by the electrostatic attraction with the membrane. In contrast, when pH was set at 4.7, the neutralized BSA freely permeated to yield a concentration of $5.02 \times 10^{-5}$ mol l$^{-1}$ after 1 h and it increased to $34.98 \times 10^{-5}$ mol l$^{-1}$ after 5 h. The positively charged BHb, on the other hand, showed negligible concentration (Fig. 6d). The efficiency of this flow-assisted separation system was obvious with superior separation selectivity (S) of 40 for BSA at pH 7.0 and 25 for BHb at pH 4.7. The reproducibility and reliability of the membranes was verified by maintaining the identical separation efficiency in the multiple MIL-53(Al)bpy$^+$-d40 nanofilter systems when implemented 2 h protein filtration and repeating 12 times (Supplementary Figure 20). To check the influence of charge in

the membrane, the separation experiment was performed in the neutral state without N-quaternization. The poor selectivity (1.14 for BHb) indicates that the separation was efficiently operated by effect of the pore and the charge. Note that no penetration of proteins (<4 nm) was experimentally proven by UV–Vis absorbance test using non-decarboxylated membrane (micropores <1 nm) (Supplementary Figure 21).

The pH-responsive excellent selectivity resulted from the heterogeneous nature of the pores in our MOF-d membrane: the ordered micropores having charged guest molecules inside provide strong electrostatic force to the target proteins, while the evolved meso- and macro-pores render the diffusive pathway to enhance the transport of the selected protein. This strategy can be extended to various applications involving large and complex molecules in conjunction with small guests.

## Discussion

We have developed a strategy for tailoring the formation of meso- and macro-pores in various MOFs via decarboxylation as a selective chemical etching process. The effectiveness of this micropore-preserving approach has been demonstrated for HKUST-1, MIL-100(Al), Eu-MOF, UiO-66, MIL-53(Al), MIL-101(Cr), and MIL-100(Fe), which includes chemically robust MOF structures. This decarboxylation etching chemistry enables engineering the spatial heterogeneity in various MOF particles, patterns, and membrane in a mild and discriminate manner. A mechanistic understanding of the decarboxylation etching chemistry was elucidated by ex-situ X-ray spectroscopies. The decarboxylation approach was utilized for UiO-66 to demonstrate that the heterogeneous pores in MOFs can be prepared on any substrate in any desired form and shape, including Si-wafer, glass slide, and flexible polyimide film, while retaining the entire integrity of the patterns and substrate. This capability of patterning and positioning meso- and macroporous MOF films on any substrates opens the door to utilizing MOFs for various energy and electronic device applications. Furthermore, we

employed this strategy to fabricate a nanofilter device by depositing heterogeneous MIL-53 film on AAO membrane as a reactive substrate. Superior separation efficiency was demonstrated for proteins of similar size by flow-assisted system and modulating the electrostatic interaction in the modified heterogeneous pores by N-quaternization and pH control. We envision that our facile and unique decarboxylation method allows cooperative utilization of heterogeneous pores for massive transfer and separation of complex and large organic, inorganic, and biological molecules.

## Methods

**Decarboxylation of MOF particles.** Decarboxylation etching of the MOFs powder for heterogeneous pores generation was conducted by mixing of 100 mg of $AgNO_3$, 100 mg of $K_2S_2O_8$, and 150 mg of MOFs in the 20 ml of ACN. The mixture was sonicated for 10 min, the mixture was transferred into 25 ml of Teflon-lined autoclave, and placed in a preheated silicon oil bath. In case of HKUST-1, the mixture was vigorously stirred at 120 °C with desired reaction time from 0, 20, 40, 60, and 90 min. In case of other MOFs [MIL-100(Al), EuMOFs, UiO-66, and MIL-53(Al)], reaction was conducted at 150 °C for 40 min and different times. When the reaction was finished, the autoclave reactor was quickly moved into ice bath to quench and prevent further decarboxylation etching. After cooled, the deionized water was introduced into resulted mixture to remove the Cu-sulfate complexes and remained catalysts from the frameworks. The decarboxylated MOFs were separated by centrifugation (4200 rpm, 12 min) for three times and finally dried in the vacuum oven at 70 °C for overnight.

**Decarboxylation for MIL-53(Al)-d40 membrane.** The etching solution was prepared by mixing 25 mg of $AgNO_3$, 25 mg of $K_2S_2O_8$ in the 20 ml of ACN, then sonicated for 30 min. The MIL-53(Al) membrane was horizontally immersed into etching solution, the mixture was transferred into 50 ml of Teflon-lined autoclave, and placed in a preheated oven. The decarboxylation of MIL-53(Al) membrane was conducted at 150 °C for 40 min. When the reaction was finished, the autoclave reactor was quickly moved into ice bath to quench and prevent further etching. After cooled down to room temperature, the substrate was rinsed with deionized water for three times. MIL-53(Al)-d40 membrane was obtained and finally dried in the vacuum oven at 70 °C for overnight. The decarboxylation method of MIL-53(Al)bpy to MIL-53(Al)bpy-d40 was exactly identical to the case of MIL-53(Al)-d40.

**N-quaternization of MIL-53(Al)bpy-d40 membrane.** To give the positive charge into/on the MIL-53(Al)bpy-d40 membrane by N-quaternization, the obtained MIL-53(Al)bpy-d40 membrane was added into 20 ml of THF in the Teflon-lined autoclave (25 ml). The autoclave was sealed and heated at 100 °C for 2 days. After the vessel cooled down to room temperature naturally, an N-quaternized MIL-53(Al)bpy-d40 membrane was obtained by washing with DMF and acetone for several times.

**Manufacturing nanofilter device.** AAO holder was fabricated by a computerized numerical control laser machine (Daewoo Heavy Industries Ltd) with poly(methyl 2-methylpropenoate). The nanofilter apparatus was devised by oppositely bonding of two same sized holder with super glue (Loctite 401). MIL-53(Al)bpy-d40 membrane was mounted into this apparatus at intermediate part. *Notice*: The commercial AAO membrane is surrounded by the polypropylene support ring. This ring should be used for bonding part. Direct contact between device and AAO membrane should be avoided because AAO membrane is fragile and will easily be cracked.

**Similar-sized protein separation.** The free-standing MIL-53(Al)bpy-d40 membrane was mounted into a home-made assembly holder. First, 10 mg of BSA (66 kDa) and 10 mg of BHb (65 kDa) were dissolved into the 3 ml of deionized water. Then, 100 μl of mixture were taken from initial solution and re-dispersed into 9.9 ml of deionized water (total concentration = 0.066 mg ml⁻¹). This solution was injected by a syringe pump (PHD 2000; Harvard Instruments, USA) into the top reservoir with 0.16 ml min⁻¹ flow rate. The separation time was varied from 1 to 5 h with 1 h of interval. Collected samples at 1–5 h were dried in 70 °C vacuum oven for overnight and re-dispersed in the excessive amount of deionized water (10 ml). The protein concentration in the bottom reservoir was determined by a NanoDrop 2000c spectrophotometer (Thermo Fisher Scientific). It is well known that the BHb in solution show two absorbance at different wavelengths, around 280 and 405 nm, while the BSA in solution exhibits one absorbance wavelength at 280 nm[30]. Therefore, BHb concentration was calculated by absorbance intensity of wavelength at 405 nm, while BSA concentration was evaluated by absorbance intensity of wavelength at 208 nm. The separation selectivity $S$ is defined as $S = F_{BSA}/F_{BHb}$ for $S_{BSA}$ at pH = 4.7 or $S = F_{BHb}/F_{BSA}$ for $S_{BHb}$ at pH = 7.0, respectively.

## Data availability

The data that support the findings of this study are available in the article or its Supplementary File, or available from the corresponding author on request.

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

## Acknowledgements

We acknowledge the National Research Foundation of Korea (NRF) grant funded by the Korean government (MSIP) (No. 2017R1A3B1023598). K.M.C. acknowledges support from Basic Science Research Program through the National Research Foundation of Korea (NRF) (No. 2016R1C1B1010781).

## Author contributions

D.-P.K. conceived the project. G.-Y.J. and K.W.G. conducted the experiments. G.-Y.J., A. K.S., U.J.R, M.-G.K. and K.M.C. contributed to analysis of data. G.-Y.J., A.K.S., K.M.C. and D.-P.K. wrote the paper.
