## [Peer Review File · Nature Communications]

Reviewers' Comments:

Reviewer #1:

Remarks to the Author:

The authors present a new method for tailoring the formation of meso- and macro-pores in various MOFs via decarboxylation. The results are interesting, and the work is publishable in NC, subjected to major revision.

- 1) The silver catalyzed decarboxylation process should be further confirmed. For example, the aryl fragment in both MOF-D and the byproduct of the etching solution can be detected by NMR spectra.
- 2) The residue of the AgNO₃ and K₂S₂O₈ in MOF-d should be analyzed to make sure the purity of the MOF-d with heterogeneous pores.
- 3) What will happen if the etching time or the etching temperature is further increased?
- 4) Some characterization of the samples should be added. For example, the PXRD of UiO-66-d40 pattern, Eu-MOF, and MIL-53(Al)bpy.
- 5) Some chemically robust MOFs such as UiO-66, Eu-MOF, MIL-100(Al) and MIL-53(Al) were selected to test the robustness of the decarboxylation approach for the preparation MOFs with heterogeneous pores. Does this approach still work for the MOFs with extremely stability, such as Cr-MOFs and Fe-MOFs?
- 6) In the last section, the adsorption studies with large biomolecules are appealing; however, there is no description regarding bovine serum albumin (BSA, 66 kDa) and bovine hemoglobin (Bhb, 65 kDa). What is the hydrodynamic dimension of biomolecules? Does it correlate with the size of the meso pores?
- 7) Characterization information of the MIL-53(Al)bpy-d40 and MIL-53(Al)dmbpy-d40 should be provided and the stability of the MIL-53(Al)dmbpy-d40 in buffer solution should be tested before it is used for protein separation experiments.
- 8) It is suggested to provide the reproducibility and reliability test of the obtained membranes.
- 9) It is still not quite clear for separation mechanism. What is the decisive factor, the electrostatic attraction or the pore size? How to characterize "The neutral Bhb should freely pass through the meso- and macro-pores in MIL-53(Al)bpy-d membrane while the negatively charged BSA would be trapped by the electrostatic attraction with the membrane"?

Reviewer #2:

Remarks to the Author:

In this communication, Kim et al. present a new strategy, called silver catalyzed decarboxylation, to control the etching of different MOFs and create macroporosity in the architecture of MOF crystals. The first study concerns the HKUST-1. HKUST-1 crystals are mixed with AgNO₃ and K₂S₂O₈ in MeCN. The authors analyzed by SEM, X-ray and N₂ adsorption analysis the evolution of the decarboxylation process. The SEM clearly shows a progressive etching of the surface of the MOF, without big changes on the X-ray powder diffraction pattern. The etching induces the formation of mesopores and macro-pores that affects the total uptake and the adsorption profile (presence of hysteresis) of the different HKUST-1 samples. Once the viability of the strategy validated, the authors extended to other MOFs (UiO-66, Eu-MOF, MIL-100 and MIL-53). In all the cases the authors observed the etching of the surface of the the crystals and changes on the adsorptions. The decarboxylation also work on MOF growth on surfaces as the authors show with different pattern of UiO-66 on silanized surfaces and MIL-53 grown on surfaces. Finally, The authors take advantage of this to prepare different membranes for protein separation application. The present study is complete, combining different characterization methods to demonstrate that the new strategy can be considered as general for different MOFs. However I have some concerns concerning different aspects of the paper. For this reason I cannot accept the publication of this paper as it is.

First; the novelty. There are some recent studies (also recent, see Kim et I. Chem. Sci 2017, 8,

6799; zhou et al.; doi:10.1038/ncomms15356) that show high control on the etching process and on the formation of mesoporosity on MOF crystals. It is true that this approach seems more general than the published ones but the added value come from surface growth and use of this etched MOFs as nanofilters that are also known areas. In my opinion, Each part (decarboxylation, surface patterning and nanofilters) are not novel enough as a communication for nat comm requires.

My other concern is about the characterization of the mesoporosity and macroporosity created by decarboxylation process. Contrariwise the authors claim, the XRPD present some differences when the decarboxylation time increase, the broadness of the peaks increase. In addition, the holes created on the surfaces and characterized by SEM seem quite big (no scale bars on the inset!) specially in view of the pore size distribution determined by BJH. The authors need to clarify this and try to correlate the explanation. Can the authors quantify the amount of carboxylic acid etched??? Furthermore, the N₂ adsorption isotherms that show an increase on the hysteresis from d20 to d40 shows a decrease from d40 to d90. Furthermore the loose of total uptake is also considerable. This part needs to be clarified.

Concerning the pattern, the strategies seem interesting. However I expect more sem images showing the UiO-66 growth on the pattern. With the size of the patterns in the photography of Figure 3 I expect more SEM images and X-ray analysis. In the other side, The MiL-53 growth on AAO is well reported by Gao et al., scientific reports, 2014, 4947. Also, the experiment with FTIC should be described and may also connected to the zeta potential of decarboxylated particles that are positive! The interaction between positive particles and COO⁻ of FTIC dextran could be enough to attract the dye to the membrane???

Finally the authors used decarboxylated mil-53 membranes for nanofiltration of proteins. I have also some concerns about this experiment. According to the authors the important interaction between the membranes and the proteins is electrostatic. In this part the authors need to clarify the different contribution between the macro-meso porosity and the positive charge of the membranes.... Did the authors performed the same experiment with a non decarboxylated membrane? Did the authors check the influence of charge of the surface on the filtration efficiency? Did the authors try to neutralize the surface of the membrane with a base after decarboxylation to see the effect of the pore and the charge?

The effect of membrane is not very clear in terms of charge effect. It seems that membranes filter positive and negative proteins but not neutral. Why?

Reviewer #3:

Remarks to the Author:

The idea proposed in the manuscript is original. With the decarboxylation of MOFs pores large enough for protein transport were formed. A strong surface charge effect switches the separation of proteins depending on pH. My concerns are

(1) The stability of the membrane after decarboxylation. Would the membrane resist a regular test in a dead end ultrafiltration cell at about 2 bar? For a few hours? Why not to measure with a similar set-up adopted by a large number of membrane-related papers?

The MOFs are known, but clearly reporting the exact chemical structure of the MOFs before and after the reaction would help to understand the process and final integrity

(2) The comparison with other similar reports (line 270) only makes sense if the measurement is done under the same conditions. According to the description of both papers, this is not the case. What is the water permeance of the MOF membrane reported here?

(3) A leak test should not be done using a dextran of 2000000 g/mol. This is an extreme molecular weight, which tends to gel and hardly would pass even defects. Is dextran with a molecular weight around 100000 g/mol available for similar measurement?

(4) Figure 3 should be changed, eliminating practically half of the images, which are not adding much information.

Response to Referees

Response to Reviewer #1

The silver catalyzed decarboxylation process should be further confirmed. For example, the aryl fragment in both MOF-D and the byproduct of the etching solution can be detected by NMR spectra.

Thanks for the Reviewer for the valuable comment. The decarboxylated powder samples (MOF-D) were determined by ATR-IR spectroscopy while the supernatant solutions after hydrothermal etching process were directly analyzed by GC-MS. Note that the complexation of the remaining organic and metal fragments in the NMR solution made difficult to give accurate information. To address these analyses, the following part was added to Figure S13-14 in the revised Supplementary Information and a text in 167-177 of p.8 in the revised manuscript.

“The remained carboxyl ligand of HKUST-1-d20, 40, 60 and 90 samples was quantitatively monitored by measuring CO_2^- stretching peak of Infrared (IR) spectrum (Figure S13a). The integrated intensities of a weak asymmetrical stretching, $\nu_{\text{as}}(\text{CO}_2^-)$, at $1400\text{--}1500\text{ cm}^{-1}$ and a relatively strong symmetrical stretching, $\nu_{\text{s}}(\text{CO}_2^-)$, at $1300\text{--}1400\text{ cm}^{-1}$ region, were progressively decreased with 7%, 11% for HKUST-1-d20, -d40, then severely reduced with 31%, 43% upon extended decarboxylation reaction to 60 min, 90 min, respectively (Figure S13). Moreover, the supernatant solutions obtained from HKUST-1-d20, 40, 60 samples were analyzed by gas chromatography mass spectroscopy (GC-MS, Figure S14). The volatile benzene peak was commonly found with acetonitrile (ACN) and dichloromethane (DCM, as activating agent). In particular, the HKUST-1-d40 and -d60 samples showed obvious presence of benzene in the solutions, which is clearly evident for decarboxylation reaction of H_3BTC (1,3,5-benzenetricarboxylic acid) ligand.”

The residue of the AgNO_3 and $\text{K}_2\text{S}_2\text{O}_8$ in MOF-d should be analyzed to make sure the purity of the MOF-d with heterogeneous pores.

As requested by the Reviewer, the residues of Ag^{2+} and K^{2+} in HKUST-1-d40 were investigated by energy-dispersive X-ray spectroscopy (EDX) in SEM as wells ICP-AES analysis. To clarify this point, the following part was added to a text in 80-82 of p.4 in the revised manuscript.

“Note that there were no residues of Ag and K ion after the post-washing process, as seen by energy-dispersive X-ray spectroscopy (EDX) in SEM (Figure S2) and ICP analysis.”

In addition, the following was added in caption of Figure S2 to the supplementary information.

“The decarboxylated samples were washed thoroughly three times with fresh DI-water and finally one time with EtOH to remove all adsorbed Ag and K ions.”

What will happen if the etching time or the etching temperature is further increased?

The etching time and the etching temperature were further increased. To clarify this point, the following part was added to a text in 82-84 of p.4 in the revised manuscript. This is because all chemical bonds within MOF were all broken by over-decarboxylation reaction, then the crystal structure was also disappeared.

“When the pore formation process was extended up to 24 hours for $170\text{ }^\circ\text{C}$, the samples lost the crystallinity and crystal shape, as observed by PXRD and SEM (Figure S3)”

Some characterization of the samples should be added. For example, the PXRD of UiO-66-d40 pattern, Eu-MOF, and MIL-53(Al)bpy.

As suggested by the Reviewer, we determined PXRD mesoporous MOFs that were mentioned from the referee. The following was added in the revised Supplementary Information and a text in 212-215 of p.10 in the revised manuscript.

“The X-ray diffraction lines of UiO-66 pattern and UiO-66-d40 pattern are well-corresponding with simulated UiO-66 diffraction. It is believed that the decarboxylation approach makes larger pores with same chemical structure and crystallinity, even in presence on any substrate.”

Some chemically robust MOFs such as UiO-66, Eu-MOF, MIL-100(Al) and MIL-53(Al) were selected to test the robustness of the decarboxylation approach for the preparation MOFs with heterogeneous pores. Does this approach still work for the MOFs with extremely stability, such as Cr-MOFs and Fe-MOFs?

Basically, the decarboxylation should be effective for all MOFs having carboxyl group because the etching reaction removes the carboxyl groups in the ligands into carbon dioxide. As requested by the Reviewer, we synthesized MIL-101(Cr) and MIL-100(Fe) and conducted their decarboxylation reaction. The following part was added to a text in 129-132 of p.6 as well as in the revised manuscript.

“The formation of mesopores into extremely stable MOFs, such as MIL-101(Cr) and MIL-100(Fe) was also confirmed by TEM and PXRD for MIL-101(Cr)-d40 (Figure S10a-c) and SEM and PXRD for MIL-100(Fe)-d40 (Figure S11a-c).”

In the last section, the adsorption studies with large biomolecules are appealing; however, there is no description regarding bovine serum albumin (BSA, 66 kDa) and bovine hemoglobin (BHb, 65 kDa). What is the hydrodynamic dimension of biomolecules? Does it correlate with the size of the meso pores?

According to the pore size distribution of the decarboxylated MOF samples and the reported hydrodynamic size of proteins, the following was added to a text at 272-274, p.14 of the revised manuscript.

“As the hydrodynamic sizes of BSA and BHb are 14 x 3.8 x 3.8 nm (ellipsoid) and 6.4 x 5.5 x 5 nm (spherical), respectively (ref.#30 *ACS Nano.*, 2013, **7**, 768), the mesopores larger than 5 nm in the nanofilter provide the pathway for the proteins passing through.”

Characterization information of the MIL-53(Al)bpy-d40 and MIL-53(Al)dmbpy-d40 should be provided and the stability of the MIL-53(Al)dmbpy-d40 in buffer solution should be tested before it is used for protein separation experiments.

As suggested by the Reviewer, both PXRD of MIL-53(Al)bpy-d40 and MIL-53(Al)dmbpy-d40 (represented as MIL-53(Al)bpy⁺-d40 in the revised manuscript) were provided to confirm the hydrolytic stability after protein separation for 24 hours (Figure S20). Further, the reliability MIL-53(Al)bpy⁺-d40 nanofilter system was also tested by maintaining the original performance in protein separation during 24 hours (Figure S20).

It is suggested to provide the reproducibility and reliability test of the obtained membranes.

The membrane performance was observed same from the samples prepared separately and remain unchanged for the extended operation time up to 24 hours (Figure S20). The following was added to a text at 289-292 in p.14 of the revised manuscript.

“The reproducibility and reliability of the membranes was verified by maintaining the identical separation efficiency in the multiple MIL-53(Al)bpy⁺-d40 nanofilter systems when implemented 2 hours protein filtration and repeating 12 times (Figure S20).”

It is still not quite clear for separation mechanism. What is the decisive factor, the electrostatic attraction or the pore size? How to characterize “The neutral BHB should freely pass through the meso- and macro-pores in MIL-53(Al)bpy-d membrane while the negatively charged BSA would be trapped by the electrostatic attraction with the membrane”?

The decisive factor for the separation is electrostatic interaction combined with mesoporosity within MOF-d film. To make clearer on the separation mechanism and process, we further updated the text at 253-263 of p.13 in the revised manuscript as below:

“Our strategy is to use the meso- and macro-pores as massive pathways for continuous flow-assisted protein movement, while allowing the surrounding micropores absorb charged molecules to give electrostatic attraction/repulsion to the proteins (Figure 5). For this purpose, the MIL-53(Al) membrane was synthesized using 2,2'-bipyridine-5,5'-dicarboxylate(bpy) followed by the decarboxylation reaction. The resulting MIL-53(Al)bpy-d40 membrane was submerged in the iodomethane solution, so that the iodomethane was supposed to attach to the pyridine site and make the micropores positively charged to give MIL-53(Al)bpy⁺-d40 membrane. This process is known as N-quaternization. The zeta potential of the MIL-53(Al)bpy⁺-d40 membrane was shifted positively after the N-quaternization (Figure S18), which clearly supports the fact that N-quaternization makes the micropores of MIL-53(Al)bpy-d membrane positively charged.”

Response to the Reviewer #2

First; the novelty. There are some recent studies (also recent, see Kim et al. Chem. Sci 2017, 8, 6799; Zhou et al.; doi:10.1038/ncomms15356) that show high control on the etching process and on the formation of mesoporosity on MOF crystals. It is true that this approach seems more general than the published ones but the added value come from surface growth and use of this etched MOFs as nanofilters that are also known areas. In my opinion, Each part (decarboxylation, surface patterning and nanofilters) are not novel enough as a communication for nat comm requires.

As the Reviewer points out, the formation of mesoporosity on the MOF crystals have been important issue and thus there are many studies focusing on breakage of metal-oxide bonds and labeled organic ligand as representatively shown in the papers that the Reviewer 2 indicated. These approaches both have intrinsic weakness: the breakage of metal-oxide bonds makes meso- and macropores at the expense of advantages (crystallinity, microporosity, robustness) of MOFs, while the breakage of labeled organic ligand is limited to the MOFs having special functional groups. Moreover, the previous methods have required the use of strong acid in the process, it caused entire damage to the used substrates of patterns, which is critical for further development of various device applications. In

our study, we employed the unique decarboxylation reaction that selectively attacks carboxyl groups of the ligands without compromising the advantage of MOFs under non-acidic and mild condition. We envision a decisive etching chemistry enables to introduce the heterogeneous pore structures in various shapes of particles, patterns, and membrane, and to further use them in the nanofilter application for the first time. We hope that the Reviewer understand these novelty and importance of our approach.

My other concern is about the characterization of the mesoporosity and macroporosity created by decarboxylation process. Contrariwise the authors claim, the XRPD present some differences when the decarboxylation time increase, the broadness of the peaks increase.

It is generally expected that the increased amount of meso- and macro-pores by decarboxylation must reduce the crystal domain size in MOFs, and the diffraction peak in PXRD becomes slightly broaden as observed. Similar phenomenon was also found in the reported work at elsewhere (ref.#13 *Chem. Eur. J.*, 2015, **21**, 18029; #14 *Nat. Comm.*, 2015, **6**, 8847)). We believe that the broadness of the peak in Figure 1 is in compliance with our claim.

In addition, the holes created on the surfaces and characterized by SEM seem quite big (no scale bars on the inset!) specially in view of the pore size distribution determined by BJH. The authors need to clarify this and try to correlate the explanation.

In general, there is obvious difference in pore size ranges that can be determined by SEM and BJH. The macropore size of HKUST-1-d20 and -d40 sample (Figure 1a and b) is in the range that both SEM observation and BET analysis coincide, while the micrometers order of pore size observed by SEM in HKUST-1-d60 (Figure 1c) is out of range by BJH analysis. Note that the scale bar is added on the inset image.

Can the authors quantify the amount of carboxylic acid etched???

The amount of carboxylic acid etched was estimated by ATR-IR spectroscopy that enabled to monitor CO_2^- stretching remained in HKUST-1-dX samples (Figure S13). To discuss this point, we added the text at 168-178 of p.8 in the revised manuscript as below:

“The remained carboxyl ligand of HKUST-1-d20, 40, 60 and 90 samples were quantitatively monitored by measuring CO_2^- stretching peak of Infrared (IR) spectrum (Figure S13a). The integrated intensities of a weak asymmetrical stretching, $\nu_{\text{as}}(\text{CO}_2^-)$, at $1400\text{--}1500\text{ cm}^{-1}$ and a relatively strong symmetrical stretching, $\nu_{\text{s}}(\text{CO}_2^-)$, at $1300\text{--}1400\text{ cm}^{-1}$ region, were progressively decreased with 7%, 11% for HKUST-1-d20, -d40, then severely reduced with 31%, 43% upon extended decarboxylation reaction to 60 min, 90 min, respectively (Figure S13). Moreover, the supernatant solutions obtained from HKUST-1-d20, 40, 60 samples were analyzed by gas chromatography mass spectroscopy (GC-MS, Figure S14). The volatile benzene peak was commonly found with ACN (acetonitrile) and dichloromethane (DCM, as activating agent). In particular, the HKUST-1-d40 and -d60 samples showed obvious presence of benzene in the solutions, which is clearly evident for decarboxylation reaction of H_3BTC (1,3,5-benzenetricarboxylic acid) ligand.”

Furthermore, the N_2 adsorption isotherms that show an increase on the hysteresis from d20 to d40 shows a decrease from d40 to d90. Furthermore the loose of total uptake is also considerable.

This part needs to be clarified.

As we described in the manuscript, the mesopores are created after 20 min of decarboxylation and the size of mesopores progressively becomes larger. As shown in SEM and BJH analysis (Figure 1c and f), the size of large pores in HKUST-1-d60 and -d90 are approaching to the scale that N₂ sorption recognizes them as open space. This is the reason that the hysteresis and total uptake is decreased in HKUST-1-d60 and -d90 samples. Similar phenomenon was also found in reported work at elsewhere (ref.#14 *Nat. Comm.*, 2015, **6**, 8847).

Concerning the pattern, the strategies seem interesting. However I expect more sem images showing the UiO-66 growth on the pattern. With the size of the patterns in the photography of Figure 3 I expect more SEM images and X-ray analysis.

PXRD analysis and SEM images for the UiO-66-d40 on the pattern are added in Figure S15 of the revised supplementary information.

In the other side, The MIL-53 growth on AAO is well reported by Gao et al., scientific reports, 2014, 4947. Also, the experiment with FTIC should be described and may also connected to the zeta potential of decarboxylated particles that are positive! The interaction between positive particles and COO⁻ of FTIC dextran could be enough to attract the dye to the membrane???

Our point is to newly generate meso- and macro-porosities by applying the decarboxylation method to the MIL-53(AI) grown on AAO for the efficient separation of biomolecules as a nanofilter. This is clearly different from the report as mentioned by the Reviewer that only grew MIL-53 membrane on AAO without formation of the mesopores. The experiment with FITC was conducted only for leak test of the nanofilter system, and there is no relevance with the zeta potential of decarboxylated particles. What the zeta potential results showed is the fact that N-quaternization makes the micropores of MIL-53(AI)bpy-d40 membrane positively charged by attaching iodomethane in the pyridine site of linkers. The charged micropores enable to make the electrostatic interaction with the charged proteins moving in the meso- and macropores.

Finally the authors used decarboxylated mil-53 membranes for nanofiltration of proteins. I have also some concerns about this experiment. According to the authors the important interaction between the membranes and the proteins is electrostatic. In this part the authors need to clarify the different contribution between the macro-meso porosity and the positive charge of the membranes....

The different contribution was already described at 253-256 of p.13 in the text of the revised manuscript as below:

“Our strategy is to use the meso- and macro-pores as massive pathways for continuous flow-assisted protein movement, while allowing the surrounding micropores absorb charged molecules to give electrostatic attraction/repulsion to the proteins (Figure 5a).”

Did the authors performed the same experiment with a non decarboxylated membrane?

As suggested, the control experiment with a non-decarboxylated membrane was conducted, the following was added to the text at 295-297 in p.15 of the revised manuscript along with Figure S20 in

the revised Supplementary Information.

“Note that no penetration of proteins (< 4 nm) was experimentally proven by UV-Vis absorbance test using non-decarboxylated membrane (micropores < 1 nm) (Figure S21).”

Did the authors check the influence of charge of the surface on the filtration efficiency? Did the authors try to neutralize the surface of the membrane with a base after decarboxylation to see the effect of the pore and the charge?

As suggested, the new experiment in neutral state was conducted. To clear this point, we added the text at 292-295 in p.15 of the revised manuscript along with Figure S21 as below:

“To check the influence of charge in the membrane, the separation experiment was performed in neutral state without N-quaternization. The poor selectivity (1.14 for BHb) indicates that the separation was efficiently operated by effect of the pore and the charge.”

The effect of membrane is not very clear in terms of charge effect. It seems that membranes filter positive and negative proteins but not neutral. Why?

As the membrane is charged after the N-quaternization process, the electrostatic attraction/repulsion force to the proteins moving through the large pores is the driving force for the successful protein separation. In case of neutral proteins, there is not enough force or interaction with the membrane so that they are not filtered out.

Response to the Reviewer #3

The stability of the membrane after decarboxylation. Would the membrane resist a regular test in a dead end ultrafiltration cell at about 2 bar? For a few hours? Why not to measure with a similar set-up adopted by a large number of membrane-related papers?

The charge-switchable membrane system of this work was manufactured by adopting the set-up that was designed for protein separation under ambient conditions as reported by many literatures (ref.#28, 30). The stability of the membrane was tested by extending the operation up to 24 hours that repeatedly conducting the protein filtration at 2 hours interval by 12 times. And it was found that the performance of the membrane remains unchanged (Figure S20). Further, the 24 hours operated MIL-53(Al)bpy⁺-d40 membrane maintained the original crystal structure as confirmed by PXRD analysis (Figure S19b), indicating high durability and stability of the system even after decarboxylation step and consecutive protein separation. The performance reliability and the structure of MIL-53(Al)bpy⁺-d40 membrane are added in Figure S20 of the revised Supplementary Information.

The MOFs are known, but clearly reporting the exact chemical structure of the MOFs before and after the reaction would help to understand the process and final integrity

In general, the change of chemical structure often evolves different crystalline phase. As we already demonstrated, the XRD data of MOF-d samples are all unchanged under optimal decarboxylation reaction for 20~60 min (Figure 1d, and Figures S6-S11). The below statement in a text at 77-80 of p.4 in the revised manuscript was slightly modified.

“The fact that the diffraction peaks of HKUST-1-d20, -d40, and -d60 all coincide (Figure 1d) is a

strong indication that the chemical structure and crystallinity remains unchanged except slight peak broadening in -d60 sample even after the formation of larger pores.”

The comparison with other similar reports (line 270) only makes sense if the measurement is done under the same conditions. According to the description of both papers, this is not the case. What is the water permeance of the MOF membrane reported here?

Authors have a difficulty to understand the comment because there is inconsistency between the line number (270) and the mentioned contents. Following the logic from the referee, it is appeared that simple comparison of our experiment to the other is inappropriate due to different measurement condition. Therefore, the comparison part was removed while the water flux of the MOF membrane was newly added in a text at 265-266 of p.14 in the revised manuscript following the reported method (ref.#30 *ACS Nano.*, 2013, **7**, 768).

“The water flux was measured around $1955 \text{ L m}^{-2} \text{ h}^{-1} \text{ bar}^{-1}$, with assumption of pressure of 1 bar.”

A leak test should not be done using a dextran of 2000000 g/mol. This is an extreme molecular weight, which tends to gel and hardly would pass even defects. Is dextran with a molecular weight around 100000 g/mol available for similar measurement?

As suggested by the Reviewer, we newly performed the leak test using a dextran of 100000 g/mol and found no leak into the bottom reservoir. The text at 266-268 of p.14 and Figure 5b are updated in the revised manuscript. The text is reproduced here:

“A leakage test was performed by pouring a fluorescence dye (100,000 MW, FITC-dextran) solution into the top reservoir.”

Figure 3 should be changed, eliminating practically half of the images, which are not adding much information.

As requested by the Reviewer, Figure 3 was simplified by rearranging the several images to Figure S14 in the revised manuscript.

Reviewers' Comments:

Reviewer #1:

Remarks to the Author:

The authors have addressed my concerns properly, and it is suitable for publication now.

Reviewer #2:

Remarks to the Author:

I consider that the revised manuscript answers the majority of my questions and doubts and can be accepted in nature communications if the editor considers that the novelty is sufficient for the journal.

Reviewer #3:

Remarks to the Author:

The authors addressed my previous comments in a satisfactory way. My recommendation is to accept the manuscript.